# A Cyclic Pentamethinium Salt Induces Cancer Cell Cytotoxicity through Mitochondrial Disintegration and Metabolic Collapse

**DOI:** 10.3390/ijms20174208

**Published:** 2019-08-28

**Authors:** Radovan Krejcir, Lucie Krcova, Pavlina Zatloukalova, Tomas Briza, Philip J. Coates, Martin Sterba, Petr Muller, Jarmila Kralova, Pavel Martasek, Vladimir Kral, Borivoj Vojtesek

**Affiliations:** 1Regional Centre for Applied Molecular Oncology, Masaryk Memorial Cancer Institute, 656 53 Brno, Czech Republic; 2Department of Pediatrics and Adolescent Medicine, First Faculty of Medicine, Charles University in Prague, Kateřinská 32, 121 08 Prague 2, Czech Republic; 3Department of Analytical Chemistry, University of Chemistry and Technology Prague, Technická 5, 166 28 Prague 6, Czech Republic; 4BIOCEV, First Faculty of Medicine, Charles University, Průmyslová 595, 252 50 Vestec, Czech Republic; 5General University Hospital, U nemocnice 2, 128 08 Prague 2, Czech Republic

**Keywords:** cancer therapy, glucose metabolism, mitochondria, autophagy

## Abstract

Cancer cells preferentially utilize glycolysis for ATP production even in aerobic conditions (the Warburg effect) and adapt mitochondrial processes to their specific needs. Recent studies indicate that altered mitochondrial activities in cancer represent an actionable target for therapy. We previously showed that salt 1-**3C**, a quinoxaline unit (with cytotoxic activity) incorporated into a meso-substituted pentamethinium salt (with mitochondrial selectivity and fluorescence properties), displayed potent cytotoxic effects in vitro and in vivo, without significant toxic effects to normal tissues. Here, we investigated the cytotoxic mechanism of salt 1-**3C** compared to its analogue, salt 1-**8C**, with an extended side carbon chain. Live cell imaging demonstrated that salt 1-**3C**, but not 1-**8C**, is rapidly incorporated into mitochondria, correlating with increased cytotoxicity of salt 1-**3C**. The accumulation in mitochondria led to their fragmentation and loss of function, accompanied by increased autophagy/mitophagy. Salt 1-**3C** preferentially activated AMP-activated kinase and inhibited mammalian target of rapamycin (mTOR) signaling pathways, sensors of cellular metabolism, but did not induce apoptosis. These data indicate that salt 1-**3C** cytotoxicity involves mitochondrial perturbation and disintegration, and such compounds are promising candidates for targeting mitochondria as a weak spot of cancer.

## 1. Introduction

Mitochondria play a crucial role in the metabolism of eukaryotic organisms. Their principal and most well-known function is to provide cells with energy in the form of ATP through oxidative respiration, although they are also important for the production and utilization of many metabolic intermediates of nucleotide synthesis and lipid metabolism, and for regulating redox balance [1]. Mitochondria are well known to exhibit altered function and activity in tumor cells and are involved in redeployment of metabolic intermediates to meet the biosynthetic requirements of proliferation and increased NADPH production to maintain cellular redox status [2,3]. Mitochondria are also a crucial component of the intrinsic pathway of apoptosis, a major mechanism of drug-induced cytotoxicity.

The development of novel modulators of apoptosis and metabolic processes is an important research area for therapeutic intervention in cancer and other diseases [4,5]. Hence, many compounds have been developed recently to selectively accumulate in mitochondria, including new experimental drugs and molecular probes that are enabling researchers to study mitochondrial processes in detail [6,7]. The most common approach for delivery of such compounds to mitochondria is the covalent linkage of a lipophilic cation to a pharmacophore. Due to the negative potential of the inner mitochondrial membrane, such positively charged compounds interact with the mitochondrial matrix and specifically accumulate in mitochondria [8]. For instance, rhodamine [9], an alkyltriphenylphosphonium moiety [10], or cyanine cations [11] have been linked to biologically active molecules to study mitochondrial activities in a wide range of physiological and pathological conditions.

Our approach for mitochondrial targeting and possible use for bioimaging and theranostics is based on the introduction of heterocyclic substitutions in the gamma position of conjugated pentamethine systems, which are in turn converted to cyclic derivatives of pentamethinium salts. Methinium salts exhibit high levels of fluorescence and have been used as molecular probes to identify specific classes of biomolecules (e.g., sulfate and heparin) [12] or cellular organelles (e.g., mitochondria) [13]. Recently, we reported the synthesis and anti-tumor properties of a novel pentamethinium salt that has high fluorescence activity and selectively localizes in mitochondria of living cells. The formerly designated compound **1** [14], herein named salt 1-**3C**, consists of the cyclic product of a quinoxaline unit incorporated into a *meso*-substituted pentamethinium salt. As such, it was expected to possess pharmacological activity typical of quinoxaline derivatives, together with inherent fluorescence and selective targeting typical of methinium salts. Indeed, compound **1** selectively localized in mitochondria of living cells and was cytotoxic to cancer cells in vitro. Initial in vivo experiments showed that compound **1** is well tolerated by mice and exhibits significant inhibitory effects on the growth of human MDA-MB-231 breast cancer cell xenografts [14]. The mechanism of action of compound **1**-induced cytotoxicity is unknown, but presumably relates to effects on mitochondrial function, where the drug is selectively localized due to its positive charge and affinity for mitochondrial phospholipid cardiolipin [14].

Here, we investigated the mechanism of cytotoxicity of salt 1-**3C**. We demonstrate that salt 1-**3C** is rapidly incorporated into mitochondria, disrupting mitochondrial structure and function, which is followed by metabolic stress and ultimately leading to death of cancer cells through a non-apoptotic mechanism. These data further support the value of targeting mitochondrial activity as a selective therapy for cancer patients.

## 2. Results

### 2.1. Chemical and Biological Properties of Salts

The chemical structures of the two pentamethinium salts tested differ only in the length of their hydrocarbon chains substituted on side heteroaromates, which consist of three (salt 1-**3C**) or eight carbons (salt 1-**8C)** (Figure 1a). Salt 1-**8C** was prepared by the same method as salt 1-**3C**, which is described and fully characterized in [14]. The Appendix A provides details of the synthesis and characterization of both salts used in this study. The maximum absorption wavelength of salt 1-**3C** is 604 nm and of salt 1-**8C** is 609 nm, and salt 1-**3C** and salt 1-**8C** have emission peaks at 632 and 635 nm, respectively (Figure 1b), allowing them to be measured using the same filter set for fluorescence detection. The extended carbon chain of salt 1-**8C** reduces its fluorescence intensity, which is still sufficient for assessment of cellular localization by fluorescence microscopy and for flow cytometric analysis (Figure 1b).

Both salts are lipophilic cations and pass readily across biological membranes due to their interaction with the hydrophobic core of membranes and the negative membrane potential. It is well known that the length of the alkyl chain influences the lipophilic properties of compounds and thereby their properties to accumulate in lipid membranes [15]. To see how both salt variants were taken up by cells in response to alteration of alkyl side-chain length, uptake was measured by flow cytometry. A-375 cells were exposed to 50 nM of salt 1-**3C** or salt 1-**8C** for 30, 60, 90, and 120 min (Figure 1c,d). Due to differences in intensity of fluorescence emitted by salts, increased fluorescence intensity was determined relative to fluorescence measured after 30 min. These experiments revealed that salt 1-**8C** enters cells more easily, showing a higher increase of cellular uptake over time.

### 2.2. Mitochondrial Localization of Salts

We generated a stable A-375-SSBP1-GFP (single-stranded DNA-binding protein, mitochondrial-green fluorescent protein) cell line derivative that expresses a GFP-tagged SSBP1 protein that is localized in mitochondria. These cells were used to verify the previous suggestion of mitochondrial localization [12,14], and to further characterize induced morphological changes. SSBP1-GFP indicates that mitochondria exist as large interconnected membrane-bound tubular networks under normal growth conditions in these cells (Figure 2a). The A-375-SSBP1-GFP cells were treated with 2000 nM salt 1-**3C** or 5000 nM salt 1-**8C** for 15 min, 4 h, and 24 h and fluorescence was measured for GFP (green) and for the salts (red). Salt 1-**3C** accumulated in mitochondria within minutes and induced fragmentation of mitochondria as early as 15 min after the treatment (Figure 2b), increasing over time. In contrast, salt 1-**8C** accumulated in vesicular structures in the cytoplasm (Figure 2c) but did not accumulate in mitochondria within 4 h.

Both salts induced formation of SSBP1-GFP-punctuate patterns indicative of mitochondrial fragmentation after 24 h (Figure 2b,c in *t* = 24 h). At this time, fragmented and shortened mitochondria were found to cluster in the perinuclear space and contained accumulated salts.

### 2.3. Salt 1-**3C** Lowers Mitochondrial Membrane Potential

To elucidate the effect of salt 1-**3C** on mitochondrial fragmentation, we further explored the mitochondrial membrane potential (MMP). MMP is an important parameter of mitochondrial function and an indicator of mitochondrial health. MMP was assessed by a fluorescent JC-1 probe that selectively enters the mitochondria. At high MMP, JC-1 forms aggregates and emits red fluorescence. In cells with low MMP, JC-1 remains in the monomeric form. Mitochondrial depolarization was determined from median values of overall distribution of red and green fluorescence of JC-1. A-375 cells were treated with 2000 nM salt 1-**3C** and 5000 nM salt 1-**8C** for 15 min, 2 h, 4 h, and 24 h. The MMP in A-375 cells treated with 2000 nM salt 1-**3C** dropped within 15 min, as indicated by the appearance of a cell population with low intensity red fluorescence, corresponding to cells that do not contain JC-1 aggregates (Figure 2d,e). The effect became more pronounced over time (Figure 2d,e), coinciding with mitochondrial fragmentation seen in Figure 2b. In contrast, mitochondrial depolarization in cells treated with salt 1-**8C** was observed only at 24 h (Figure 2d), confirming the weaker effect on mitochondrial function.

### 2.4. Salt 1-**3C** Leads to Altered Autophagy Processes

Mitochondrial fragmentation could stimulate mitophagy as a specific removal mechanism of damaged mitochondria through autophagy. Moreover, a characteristic feature during mitophagic turnover is accumulation of fragmented and shortened mitochondria in the perinuclear space [16], as we had observed. To investigate whether autophagy/mitophagy are involved in the action of both salts, changes in microtubule-associated protein light chain 3 (LC3) and the autophagy receptor sequestosome 1 (p62/SQSTM1) were investigated by immunoblotting. Cytoplasmic LC3 is processed and recruited to autophagosomal membranes and is frequently used to monitor autophagy. p62/SQSTM1 is an adaptor protein that attaches autophagic cargoes to the autophagic membrane through interaction with LC3 [17]. Autophagy is a dynamic process that also includes autophagosome maturation and degradation in lysosomes, leading to decreased levels of p62/SQSTM1 and LC3 during the late stage of autophagy. To investigate whether autophagic flux is increased in response to salt 1-**3C** or salt 1-**8C**, LC3 levels were assessed in relation to p62/SQSTM1. A-375 cells were exposed to 100, 500, and 2000 nM salt 1-**3C** and to 500, 2000, and 5000 nM salt 1-**8C** (Figure 3a,b,c). Salt 1-**3C** did not lead to altered levels of LC3 or p62/SQSTM1 at early time points (before 12 h, data not shown). High concentrations of salt 1-**3C** led to an increase of LC3 after 12 h and decreased levels at later times, whereas lower concentrations increased levels of LC3 only at later times (Figure 3a). With different kinetics and dose-dependency, a similar pattern was seen for p62/SQSTM1, where lower salt 1-**3C** concentrations caused an increase in p62/SQSTM1 levels at 24 h that reduced over time, whereas high concentrations of salt 1-**3C** led to a reduction of p62/SQSTM1 within 24 h (Figure 3b). Salt 1-**8C** did not lead to consistent changes in LC3 or p62/SQSTM1 (Figure 3a,b). Metformin served as a positive control for assessment of autophagy [18] and showed a dose- and time-dependent reduction in LC3 and p62/SQSTM1 (Figure 3a,b).

To further investigate autophagic flux alterations caused by the compounds, A-375 cells were treated with 100 nM bafilomycin A1 (bafA1) for 1, 2, 3, and 4 h. BafA1 is a V-ATPase inhibitor that increases lysosomal pH and thereby blocks autophagic flux by inhibiting autophagosome–lysosome fusion, allowing quantification of the total levels of generated LC3. Figure 3d shows a time-dependent increase in LC3 in A-375 cells treated with bafA1 alone. Treatment with 2000 nM salt 1-**3C** for 24 h followed by 100 nM bafA1 for 4 h resulted in an increase of both LC3 and p62/SQSTM1, which corresponds to the total level of autophagosomes generated after treatment with salt 1-**3C** (Figure 3e).

Stress-induced mitochondrial fragmentation activates mitochondrial quality control systems, which involve synthesis of new mitochondria, mitochondrial fusion and fission, and elimination of damaged mitochondria. As mentioned above, mitophagy specifically recognizes damaged mitochondria and delivers them to lysosomes for degradation [19]. To evaluate whether salt 1-**3C** induces removal of fragmented mitochondria by autophagy, LC3 immunostaining was performed after treatment with 2000 nM salt 1-**3C** for 24 h. LC3 positive puncta were observed throughout the cytoplasm, including regions not occupied with damaged mitochondria (Figure 3f). There were no overlaps between signals of LC3 (green) and mitochondria with accumulated salt 1-**3C** (red). These findings indicate that autophagy was activated as a survival process, helping cells to balance the impact of damaged mitochondria and implying that affected cells are unable to efficiently process all of the damaged mitochondria by removal and recycling through mitophagy.

### 2.5. Salts 1-3C and 1-8C do not Induce Apoptosis

The mitochondrial fission machinery can actively participate in the regulation of apoptosis [20,21]. Mitochondrial fragmentation influences remodeling of cristae which can subsequently lead to the release of cytochrome *c* or other pro-apoptotic factors that ultimately trigger caspase activation [22,23]. Proteolytic cleavage of poly(ADP-ribose) polymerase (PARP) and caspase-3 (casp-3) correlates with induction of apoptosis. Cleavage of PARP produces an 89 kDa C-terminal fragment during apoptosis. Activation of casp-3 requires proteolytic processing of its inactive proenzyme into activated 17 and 12 kDa fragments. A-375 cells treated with cisplatin were used as positive control, showing PARP and casp-3 cleavage. However, treatment with either salt did not cause cleavage of PARP or casp-3 after 24 or 48 h (Figure 4a,b).

### 2.6. Impact on Mitochondrial Bioenergetics through AMP-Activated Protein Kinase–Mammalian Target of Rapamycin (AMPK–mTOR) Signaling

Alterations of mitochondrial morphology may affect bioenergetic status and consequently metabolic rate. Stresses that deplete cellular ATP and increase the ADP:ATP ratio activate AMP-activated protein kinase (AMPK). Activation of AMPK involves phosphorylation of Thr172, which was detected by Western blotting (Figure 5a). Metformin was used as a positive control [24]. While the total level of AMPK remained unchanged after treatment with both salts, (Thr172) phosphorylated AMPK (P-AMPK) showed a dose- and time-dependent increase, more evident after treatment with salt 1-**3C** (Figure 5a), in which increased P-AMPK levels were observed as early as 24 h after treatment with 500 nM salt 1-**3C,** with 100 nM also increasing P-AMPK at 48 h. The activation of AMPK by salt 1-**8C** was detected only after 48 h and only at the highest concentration tested (5000 nM).

AMPK is an antagonist of the mammalian target of rapamycin (mTOR), a master regulator of cell metabolism, and inhibition of mTOR activates autophagy/mitophagy [25]. A downstream target of mTOR is p70 S6 kinase (p70S6K), which is controlled by phosphorylation and subsequently phosphorylates other substrates that promote protein synthesis [26]. Exposure of A-375 cells to salt 1-**3C** reduced the levels of (Ser434) phosphorylated p70S6K (P-p70S6K) in a dose-dependent manner. Once again, a delay in dephosphorylation of p70S6K was observed after treatment with salt 1-**8C** (48 h) compared to salt 1-**3C** (Figure 5b).

To study the impact of salts on mitochondrial function, oxygen consumption rate (OCR) was measured as a direct indicator of oxidative respiration. The fluorescent probe, MitoXpressXtra, is quenched by oxygen and the fluorescence signal is therefore proportional to the OCR. The mitochondrial uncoupler, carbonyl cyanide-p-triflouromethoxyphenyl-hydrazone (FCCP), which depolarizes the mitochondrial membrane and induces maximal respiration as mitochondria attempt to restore their membrane potential [27], was used as the positive control and showed the expected increase in fluorescence signal. In contrast, antimycin A, an inhibitor of mitochondrial respiration, was used as a negative control and decreased the fluorescence signal (Figure 6).

Salt 1-**3C** decreased the OCR in a dose-and time-dependent manner, seen within 4 h of treatment (Figure 6a). Higher concentrations (2.5–5 µM) completely suppressed OCR to that of cells treated with antimycin A. The effect of salt 1-**3C** on OCR was more obvious after 24 h of treatment (Figure 6b). A weak impact of salt 1-**8C** on OCR was observed after 24 h at higher concentrations (Figure 6d).

### 2.7. Glucose and Pyruvate Deprivation Influence Cytotoxicity

A-375 cells were cultured in DMEM with different concentrations of glucose and pyruvate for 72 h. Salt 1-**3C** induced a dose-dependent decrease in cell viability for both glucose variants, with a more progressive decrease for cells grown in low glucose (LG) (Figure 7a). Cells cultured in LG were also more sensitive to pyruvate deprivation than cells grown in standard medium (high glucose; HG) (Figure 7a). Salt 1-**8C** showed lower cytotoxicity compared with salt 1-**3C**. The viability of A-375 cells in HG medium was comparable with untreated cells, even at 5000 nM salt 1-**8C** (Figure 7b) and the IC_50_ for salt 1-**8C** was established only for A-375 in the LG medium with or without pyruvate (4300 or 4600 nM, respectively) (Figure 7c). These results indicate that the cytotoxicity of both salts is influenced by glucose availability and that pyruvate supplementation aids cell survival when glucose is limiting, but not when supraphysiological levels of glucose are available.

## 3. Discussion

We prepared benzothiazolium cyclic pentamethinium salts (salt 1-**3C** and salt 1-**8C**), where lipophilicity was tuned by length of the alkyl chain on the side heteroaromates. Both salts are lipophilic cations that preferentially incorporate into membranous organelles through insertion of the alkyl chain into the phospholipid bilayer. Interestingly, lipophilic cations can pass through membranes and accumulate in negatively-charged compartments such as the mitochondrial matrix. Currently, fluorescent lipophilic cations such as rhodamine, JC-1, and MitoTracker® are routinely used to visualize mitochondria within cells [28]. We chose salt 1-**3C** as a pilot model for mechanistic studies, which was developed and tested in our institute a few years ago [14]. This compound exhibits 20- to 400-fold higher cytotoxicity for a range of cancer cells compared to non-transformed cells and has no discernible effects on normal cell populations in murine xenograft experiments [14]. As an appropriate molecule for comparison with salt 1-**3C**, we synthesized a compound with the same structural motif, but with increased lipophilicity, which can significantly affect its transport into the cell, its localization, and its toxicity. We chose a pentamethinium salt with two octyl chains C8 on the side benzothiazole units as the model (salt 1-**8C**). The length of the C8 chain proved to be almost ideal for our studies, as longer chains caused complications in transferring to the aqueous phase and made the compound essentially unusable for our further cellular studies.

Mitochondria are potential targets for cancer therapy. Accumulation of salt 1-**3C** is dependent on metabolic activity, with little or no cellular uptake observed at 4 °C [12], indicating a potential application for treatment of cells with increased metabolic activity, which is typical for tumor cells. Moreover, salt 1-**3C** displays in vitro cytotoxic effects in a wide range of cancer cell lines and in vivo suppression of tumor xenograft growth in mice [12], although the mechanism of action has not been studied previously.

Mitochondria are surrounded by a double-membrane system, consisting of inner and outer mitochondrial membranes separated by an intermembrane space. The outer mitochondrial membrane is highly porous and relatively permeable for ions and small and uncharged molecules. The inner membrane, with its membrane potential (Δψ) and pH gradient, poses a diffusion barrier for all molecules. We found that salt 1-**3C** exhibits mitochondrial selectivity. Accumulation of salt 1-**8C** in mitochondria is less marked, correlating with the lower cytotoxicity of salt 1-**8C** than salt 1-**3C** and suggesting that the processes leading to cell death have been delayed. The altered location could be due to the extended alkyl side-chain of salt 1-**8C,** which may obstruct effective penetration through the inner mitochondrial membrane, or it may have lower affinity to mitochondrial target(s) common for both salts.

Given the selectivity of salt 1-**3C** for mitochondria, we hypothesized that its cytotoxicity could be related to altered mitochondrial function. In keeping with this notion, we showed that salt 1-**3C** rapidly induced mitochondrial fragmentation and led to accumulation of small punctate mitochondria in the perinuclear space. Mitochondria exist as dynamic networks that are continually modified through the combined actions of mitochondrial fragmentation/fission and fusion that continuously remodel mitochondrial morphology and number [28,29], which are essential to maintain mitochondrial integrity and function and play important roles in regulating stress-related processes [30]. Salt 1-**3C** disrupted energy metabolism and activated the energy sensor AMPK that regulates metabolism by switching on catabolic pathways to generate ATP and switching off processes that consume ATP through direct phosphorylation of downstream substrates. We also demonstrated that salt 1-**3C** inhibits mTOR signaling, which allows cells to survive by regulating processes that exploit nutrients and sources of energy. Thus, these data indicate a cellular response that reflects overall mitochondrial conditions. We suggested that salt 1-**3C** mediates autophagy in response to altered mitochondrial structure and/or function. Autophagy is a protective process, defending cells from death by recycling cytoplasmic components and organelles to replenish essential metabolites. Autophagy also mediates self-destructive collapse of cells once large proportions of the cytosol and organelles are destroyed due to long-term stress conditions. Activation of autophagy may also be a primary response to stress that subsequently triggers either apoptosis or necrotic cell death [29]. Signaling pathways leading to these different types of cell death are interconnected and can be activated simultaneously or operate in parallel [30]. Western blotting confirmed that salt 1-**3C** primarily induces autophagy, with no evidence for apoptotic cell death. Microscopic examination also demonstrated that apoptosis does not contribute to the cytotoxic effect of salt 1-**3C** (no nuclear condensation was observed). Generally, autophagosomes are formed in the cytoplasm and carried towards lysosomes, which are predominantly localized in the perinuclear space [31] and are the location of late occurring mitochondrial fragments after salt 1-**3C** treatment. These data suggest that salt 1-**3C** disrupts mitophagic flux, because aggresomes with salt 1-**3C** remain in the perinuclear space, implying an inability to fuse with lysosomes for destruction and recycling, evidenced by the effect of bafilomycin 1A.

The glycolytic phenotype is an attribute of almost all types of cancer cells and a well-known adaptation process is to increase aerobic glycolysis to generate ATP, the Warburg effect. Although respiration occurs at low intensity in tumor cells, it is essential to maintain the mitochondrial membrane potential and thus cell survival and division. Besides ATP generation, cancer cells utilize almost 10% of their glucose for biosynthetic pathways to cover the requirements of basal metabolism and to support cell proliferation [32]. Glucose metabolites are redirected into nucleotide or amino acid biosynthetic pathways, as well as to the pentose phosphate pathway. This metabolic pathway is directly linked to glycolysis and produces NADPH necessary for the synthesis of fatty acids and lipids and protects the cell from free radicals. Moreover, glycolysis rate can be limited by the amount of cytosolic NAD+ (oxidized form of NADH), which is reduced to NADH in one of the steps. NADH is usually oxidized back to NAD+ during mitochondrial respiration. In cancer cells, deficiency of NAD+ is often compensated by increased production of lactate dehydrogenase, which converts pyruvate in the cytosol to lactate, during which NAD+ is regenerated. However, this increases the levels of lactate, which could itself decrease the rate of glycolysis, and lactate is therefore released from cancer cells to maintain pH. Pyruvate also occupies a central position in the metabolism of substrates or intermediates exploitable in biosynthetic pathways or in activation of pathways generating reducing agents.

We demonstrated that salt 1-**3C** disrupts mitochondrial function, reflected by a decrease in membrane potential and oxidative phosphorylation (OXPHOS). We also found that the availability of glucose and pyruvate influenced the cytotoxic effect. In the future, it will be important to investigate other mitochondrial parameters, such as the effects on cellular ATP concentrations. The analysis of cytochrome c release, addition of caspase inhibitors, and more detailed analysis of the time course of changes in autophagy-related proteins will also be useful to further study the exact mechanisms of cell death and the autophagic engulfment of mitochondria, including the use of isolated mitochondria to study the direct effects of salt 1-**3C** on mitochondria without the interfering effects of autophagosomes. Bearing in mind that salt 1-**3C** is cytotoxic for cancer cells but not normal cells [14], whether cancer cell death is due to the alterations in OXPHOS and consequent reduction in ATP generation, or is due to OXPHOS-independent effects on mitochondria remains to be determined. Similarly, a more extensive study of normal versus cancer cells and relationships with proliferating versus non-proliferating cells are now warranted from our studies of salt 1-**3C**. A similar effect was observed for salt 1-**8C**, but was delayed and shifted to higher concentrations, presumably reflecting its lower selective mitochondrial uptake. From our findings, we conclude that mitochondrial damage caused by salt 1-**3C** leads to overall metabolic collapse, particularly when glucose (or pyruvate) are depleted, and eventually results in autophagy-related but apoptotic-independent cell death (Figure 8). In light of our findings, the relationships between respiratory actions, metabolic actions and cancer cell cytotoxicity due to mitochondrial targeting need further investigation, particularly with regard to the potential for cancer therapy.

## 4. Materials and Methods

### 4.1. Synthesis of Salt 1-**3C** and Salt 1-**8C**

The methods used for synthesis of salt 1-**3C** and salt 1-**8C** and their characterization by 1H, 13C, and high resolution mass spectrometry are provided in the Appendix A.

### 4.2. Fluorescence Spectra

Fluorescence measurements were performed on a FluoroMate FS 2 spectrometer (SCINCO Co. Ltd., Seoul, Korea) using a 1 cm poly (methyl methacrylate)cell. The tested compounds were dissolved in DMSO to 62.5 nM final concentration.

### 4.3. Cell Culture and Reagents

A-375 human malignant melanoma cells were obtained from ATCC (ATCC®, Manassas, VA, USA-CRL-1619™) and maintained in Dulbecco’s modified Eagle’s medium (DMEM) (Sigma-Aldrich, St. Louis, MO, USA) containing 10% fetal bovine serum, 1% pyruvate, and penicillin/streptomycin at 37 °C in a humidified atmosphere with 5% CO_2_. Cells were routinely grown and maintained in DMEM with high-glucose (4500 mg/L), or in low-glucose (1000 mg/L) containing all supplements when indicated. In some experiments, cells were grown in medium without pyruvate. Cells were grown to 70%–80% confluence prior to treatment with metformin and cisplatin (Sigma-Aldrich). As a model for comparison to salt 1-**3C** we used the more lipophilic salt 1-**8C**, which is substituted with octyl chains on side heteroaromates (Figure 1a). Based on preliminary data of different cytotoxicities, lipophilicity, and the optical properties of the two compounds, different doses of each compound were used to compare their effects.

### 4.4. Flow Cytometry Analysis of Uptake Kinetics and Mitochondrial Membrane Potential

A-375 cells were seeded in 12-well plates using 100,000 cells per well and left to adhere for 24 h. Cells were treated with 50 nM salt 1-**3C** or salt 1-**8C** for 30, 60, 90, and 120 min. Cells were harvested with accutase solution (Sigma-Aldrich), washed twice with phosphate buffered saline (PBS), and resuspended in PBS. Uptake of both compounds was measured by the fluorescent properties of the compounds detected in the channels for APC (allophycocyanin) (Ex-Max 650 nm/Em-Max 660 nm). The fluorescent signals were evaluated by BD FACSVerse™ (BD Biosciences, Franklin Lakes, NJ, USA) using Cell Quest software (BD Biosciences). All samples were analyzed in triplicate.

Mitochondrial membrane potential was analyzed using JC-1 dye (Thermo Fisher Scientific, Waltham, MA, USA). A-375 cells at a concentration of 250,000 cells per well were cultured overnight in 6-well plates. Cells were treated with 2000 nM salt 1-**3C** and 5000 nM salt 1-**8C** for 15 min, 2 h, 4 h, and 24 h or were untreated (time 0). Cells were harvested with accutase solution, washed twice with PBS, resuspended in PBS with 5 µg/ml JC-1, and measured by FACSAria^TM^ (BD Biosciences). Data were analyzed in triplicate by FCS Express version 4.0 (De Novo Software, Glendale, CA, USA).

### 4.5. Generation of A-375-SSBP1-GFP and Live Cell Imaging

An A-375 stable cell line expressing mitochondrial single-stranded DNA-binding protein (SSBP1) fused with green fluorescence protein (GFP) (A-375-SSBP1-GFP) was generated. The full coding sequence of *SSBP1* was amplified by PCR from cDNA using primers: GTCTCTACGCTGCTTGGTCT and ATTCTTGTCCTTGGAAGCCA. *SSBP1* was cloned into lentiviral plasmid with C-terminal GFP and a clone with the highest proportion of GFP-positive signal without limiting cell growth was selected. A-375-SSBP1-GFP cells were plated in µ-Slide 8 Wells (ibidi GmbH, Gräfelfing, Germany) at 10,000 cells per well and left to adhere for 24 h.

Cells were also exposed to 2000 nM salt 1-**3C** or 5000 nM salt 1-**8C** for 15 min, 4 h, and 24 h, and cells were imaged using a live cell imaging microscope (60× objective) at 37 °C, 95% humidity, and 5% CO_2_ (Nikon, Tokyo, Japan).

### 4.6. Western Blotting

Sodium dodecyl sulphate-polyacrylamide gel electrophoresis and immunoblotting were performed as described previously [33]. Chemiluminescent signals were developed using ECL and visualized with GeneTools (Syngene, Bangalore, India). Primary antibodies against PARP1 (C2-10), β-actin, SQSTM1 (A-6), phospho-p70 S6 kinase (Ser434), p70 S6 kinase (Santa Cruz Biotechnology, Dallas, Texas, USA), caspase-3, AMPK, phospho-AMPK (Thr172), and LC3 (Novus Biologicals, Centennial, CO, USA) were used. Secondary antibodies were HRP-conjugated swine anti-rabbit and HRP-conjugated rabbit anti-mouse (Agilent, Santa Clara, CA, USA).

### 4.7. Oxygen Consumption Assay

Real-time measurement of extracellular oxygen consumption was assessed using MitoXpressXtra Oxygen Consumption Assay (Agilent). Cells were seeded into 96-well plates at 60,000 cells per well, incubated for 24 h at 37 °C/5% CO_2_ and subsequently treated with 300–5000 nM salt 1-**3C** or salt **1-8C** for 4 and 24 h. Growth medium was replaced with fresh culture medium containing reconstituted MitoXpress reagent and each well was sealed from the air supply by adding two drops of mineral oil. Fluorescence measurements were taken at excitation 380 nm and emission 650 nm after 120 min using a fluorescence plate reader (Tecan Infinite M1000 Pro, Tecan Group Ltd., Männedorf, Switzerland). MitoXpressXtra reagent is quenched by O_2_, through molecular collision, thus the amount of fluorescence signal is inversely proportional to the amount of extracellular O_2_ in the sample. Increases in fluorescence are indicative of oxygen consumption. Antimycin A (500 nM) (Sigma-Aldrich), which inhibits mitochondrial electron transport between cytochromes b and c and abolishes oxygen consumption, was used as a negative control. Carbonyl cyanide-p-triflouromethoxyphenyl-hydrazone (FCCP) (Sigma-Aldrich) (2000 nM) uncouples oxygen consumption from ATP production and raises OCR to a maximal value and was used as a positive control. Results were normalized to cell number. Each sample was measured in four technical replicates.

### 4.8. Immunofluorescence Microscopy

A-375 cells were fixed with 4% paraformaldehyde for 20 min at 37 °C and permeabilized with 0.1% Triton X-100 for 5 min at room temperature. Cells were blocked with 3% bovine serum albumin (BSA) for 1 h and then incubated overnight with the primary antibody. Next day, coverslips were washed and probed for 1 h with Alexa Fluor 488 goat anti-rabbit IgG (Abcam, Cambridge, United Kingdom). After washing, the cells were incubated with DAPI for 5 min. Coverslips were mounted with Vectashield and captured using an Olympus BX41 microscope, images were analyzed with CellSens software (Olympus, Shinjuku, Japan).

### 4.9. Viability/Metabolic Activity Assay

Cell viability was determined using Resazurin sodium salt (Sigma-Aldrich). A-375 cells were seeded into 96-well plates using 10,000 cells per well and left to adhere for 24 h. The experiments were performed in DMEM containing low glucose (1000 mg/L, LG medium) or high glucose (4500 mg/L, HG medium) supplemented with or without 1% pyruvate. Salt 1-**3C** and salt 1-**8C** concentrations were varied from 0 to 5000 nM. Resazurin was added at a final concentration of 100 mg/L and incubated for 3 h at 37 °C. Fluorescence was measured at 530 nm excitation and 590 nm emission using a fluorescence microplate reader (Tecan Infinite M1000 Pro). The concentration of salt required to achieve 50% growth inhibition (IC_50_) was analyzed after 72 h. Each sample was measured in four technical replicates.

## Figures and Tables

**Figure 1 ijms-20-04208-f001:**
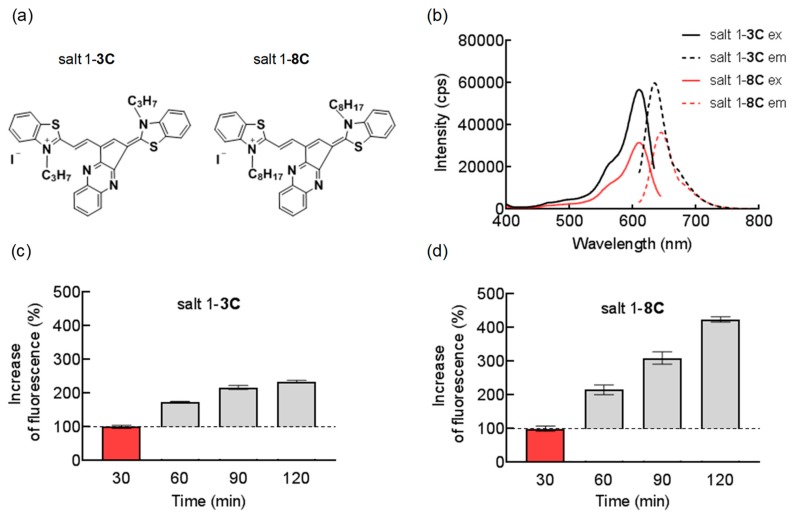
The 1-**8C** salt easily permeates cell membranes resulting in higher intracellular concentration. (**a**) Chemical structure of salt 1-**3C** and salt 1-**8C**. (**b**) Fluorescence spectra of salt 1-**3C** and salt 1-**8C** measured in DMSO (ex, excitation; em, emission). Uptake of 50 nM (**c**) salt 1-**3C** and (**d**) salt 1-**8C** measured after 30, 60, 90, and 120 min by flow cytometry. Increase of fluorescence intensity was determined relative to fluorescence measured after 30 min (red column). Emitted fluorescence corresponds to the amount of salts accumulated inside the cells.

**Figure 2 ijms-20-04208-f002:**
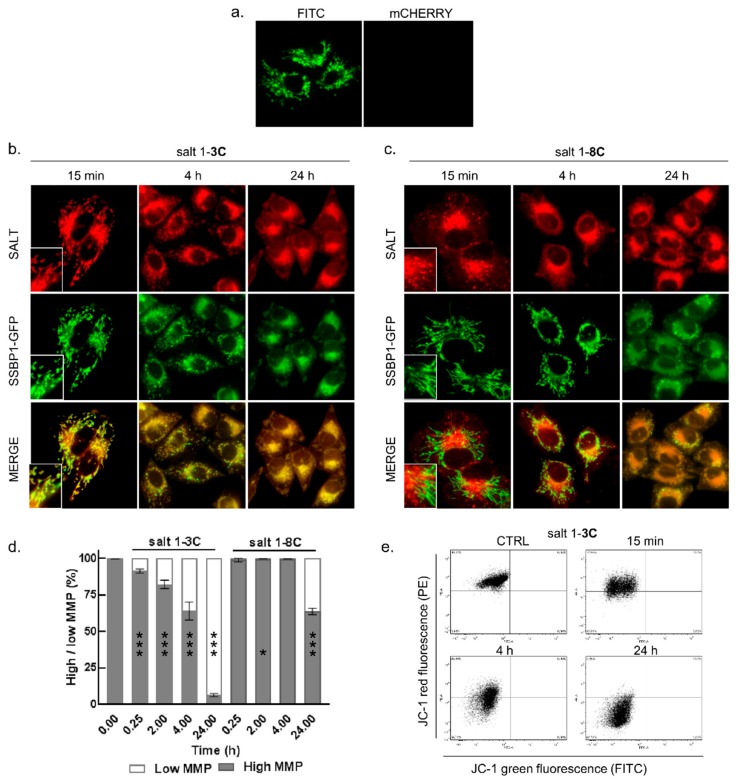
Mitochondrial accumulation and morphology changes after treatment with salt 1-**3C** and salt 1-**8C** and measurement of mitochondrial membrane potential. (**a**) A-375-SSBP1-GFP (single-stranded DNA-binding protein, mitochondrial-green fluorescent protein) cells detected in green (FITC, *λ*em = 525 nM) and red (mCherry, *λ*em = 610 nM) channels. Fluorescence microscopy images of stable A-375-SSBP1-GFP cells (600x magnification) after treatment with 2000 nM salt 1-**3C** (**b**) and 5000 nM salt 1-**8C** (**c**) for the indicated times (salt 1-**3C** and salt 1-**8C** in red and mitochondrial SSBP1-GFP in green). Insets show additional software magnification (3×). (**d**) Flow cytometry analysis of JC-1 (mitochondrial membrane potential (MMP)). Distribution of low (white) and high (grey) MMP in A-375 cells treated with 2000 nM salt 1-**3C** and 5000 nM salt 1-**8C** at the indicated times. The results represent the mean ± SD of technical triplicates, each with 10,000 counted cells; * *p* < 0.05, ** *p* < 0.01, *** *p* < 0.001. (**e**) Flow cytometry analysis of JC-1 in A-375 cells treated with 2000 nM salt 1-**3C** for 15 min, 4 h, and 24 h, and for untreated cells (CTRL). Upper and lower left quadrants represent aggregates and monomers of JC-1, respectively.

**Figure 3 ijms-20-04208-f003:**
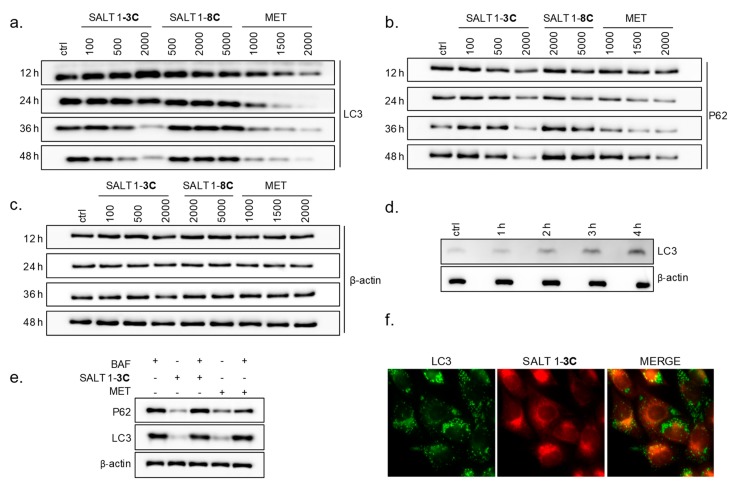
Mitochondrial fragmentation is accompanied by autophagy. Western blot of lysates prepared from A-375 cells treated with different concentrations of salt 1-**3C** (100, 500, and 2000 nM) or salt 1-**8C** (500, 2000, and 5000 nM) for different times (12, 24, 36, and 48 h). Metformin (MET) served as positive control (1000, 1500, and 2000 µM for 24 h). Microtubule-associated protein light chain 3 (LC3), autophagy receptor sequestosome 1 (p62/SQSTM1) (P62), and β-actin have apparent molecular weights on Western blots of 15, 60, and 42 kDa, respectively. (**a**) Detection of LC3. (**b**) Detection of P62. (**c**) Detection of β-actin as loading control. (**d**) Detection of LC3 and β-actin levels for A-375 cells treated with 100 nM bafilomycin 1A (BAF) for 1, 2, 3, and 4 h. (**e**) Detection of LC3, P62, and β-actin for A-375 cells treated with only BAF (100 nM for 4 h), only salt 1-**3C** (2000 nM for 28 h), and only MET (2000 µM for 28 h). In two variants, A-375 cells were pre-treated with 2000 nM MET or 2000 nM salt 1-**3C** for 24 h and afterwards treated with 100 nM BAF for 4 h. (**f**) A-375 cells treated with 2000 nM salt 1-**3C** (red) for 24 h and stained with anti-LC3 (green) (600× magnification).

**Figure 4 ijms-20-04208-f004:**
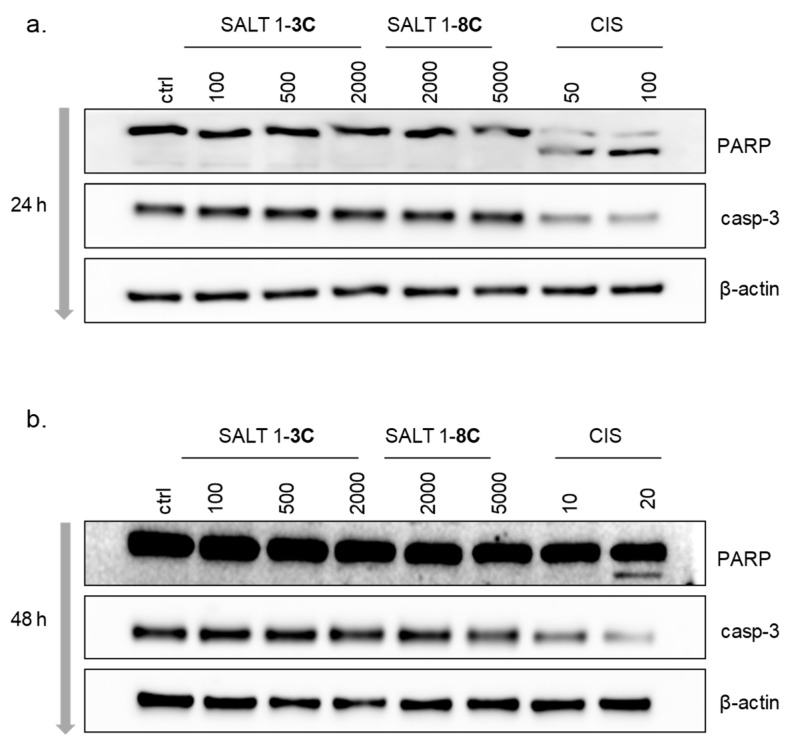
1-**3C** and 1-**8C** do not induce apoptosis. Western blot of lysates prepared from A-375 cells treated with different concentration of salt 1-**3C** (100, 500, and 2000 nM) and salt 1-**8C** (2000 and 5000 nM) for (**a**) 24 h and (**b**) 48 h. Cisplatin (CIS) (50 and 100 µM for 24 h, 10 and 20 µM for 48 h) served as the positive control. Bands of poly(ADP-ribose) polymerase (PARP)/cleaved PARP, caspase-3 (casp-3), and β-actin were detected at 116/89 kDa, 30 kDa, and 42 kDa, respectively. β-actin served as the loading control.

**Figure 5 ijms-20-04208-f005:**
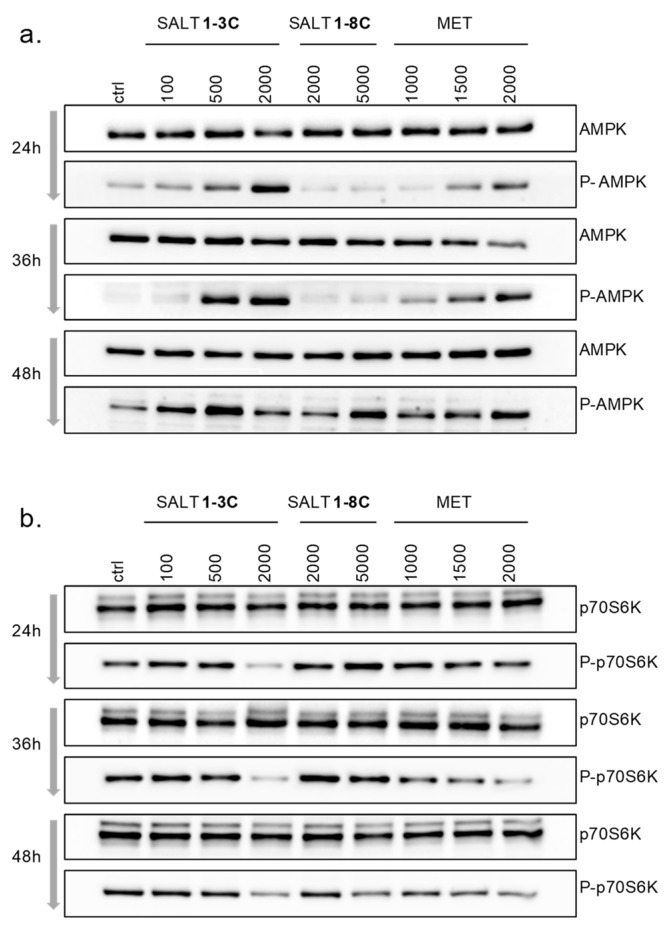
Salts 1-**3C** and 1-**8C** induce metabolic stress. Western blots of lysates prepared from A-375 cells treated with the indicated concentrations (nM) of salt 1-**3C**, salt 1-**8C,** or metformin for 24, 36, and 48 h. Metformin (MET) served as the positive control. (**a**) Detection of AMP-activated protein kinase (AMPK) and (Thr172) phosphorylated AMPK (P-AMPK). (**b**) Detection of p70S6K and (Ser434) phosphorylated p70S6K (P-p70S6K).

**Figure 6 ijms-20-04208-f006:**
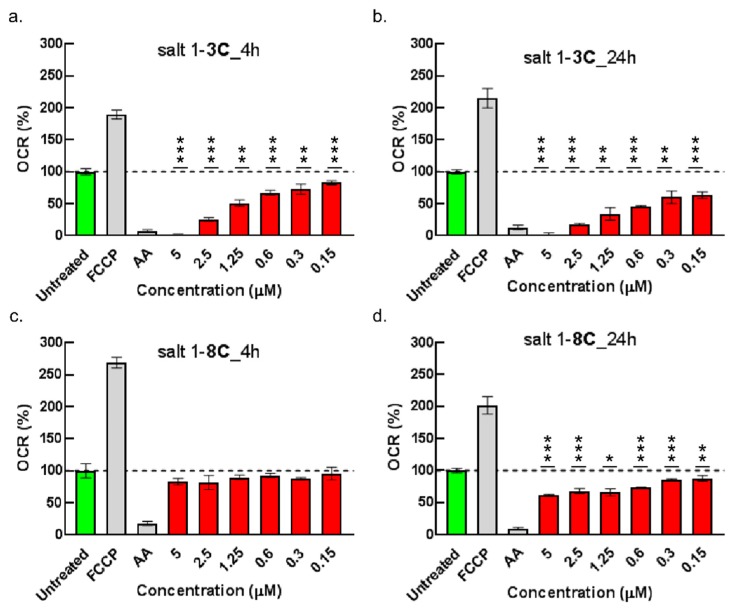
Salts influence oxygen consumption. A-375 cells were treated with the indicated concentrations of salt 1-**3C** (**a**,**b**) and salt 1-**8C** (**c**,**d**) for 4 h (**a**,**c**) and 24 h (**b**,**d**). Oxidative respiration was measured using MitoXpressXtra Oxygen Consumption Assay. Control cells are indicated in green. Carbonyl cyanide-p-triflouromethoxyphenyl-hydrazone (FCCP) (2 µM) and antimycin (AA) (500 µM) were used as positive and negative controls, indicated in grey. Test compounds are indicated in red. The results represent the mean ± SD of technical triplicates; * *p* < 0.05, ** *p* < 0.01, *** *p* < 0.001.

**Figure 7 ijms-20-04208-f007:**
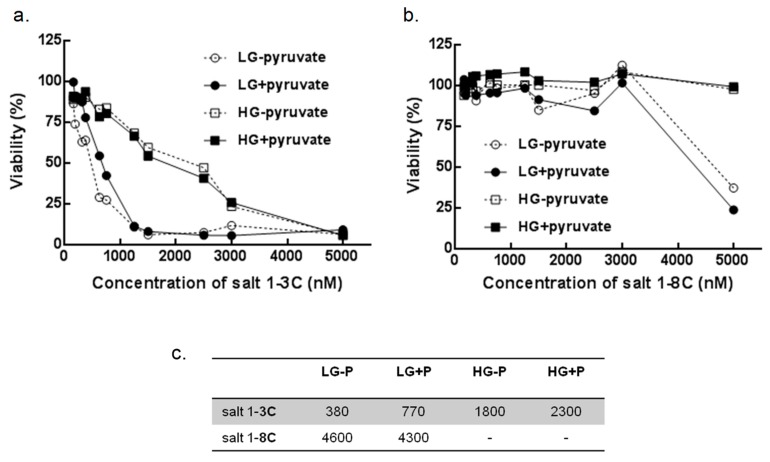
Viability depends on glucose and pyruvate availability. Resazurin viability assay for A-375 cells treated with salt 1-**3C** (**a**) or salt1-**8C** (**b**) at the indicated concentrations and cultured in Dulbecco’s modified Eagle’s medium (DMEM) with low (LG) or high (HG) glucose and with (+P) or without (−P) pyruvate for 72 h. (**c**) IC_50_ (nM) determined from **A**. Each sample was measured in five replicates.

**Figure 8 ijms-20-04208-f008:**
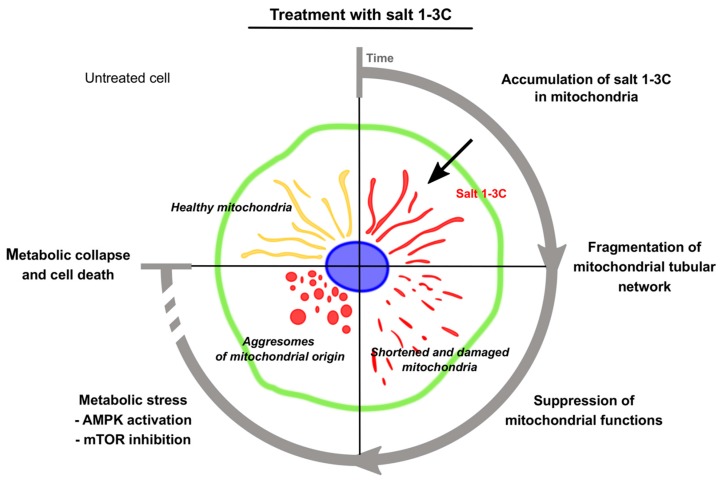
Mechanism of cytotoxic action of salt 1-**3C** leading to metabolic collapse and cell death. Salt 1-**3C** rapidly permeates cell membranes, accumulates in mitochondria, and induces mitochondrial fragmentation as early as 15 min after treatment, becoming more pronounced over time. Alterations of mitochondrial morphology are accompanied by suppression of oxygen consumption. Salt 1-**3C** accumulates in aggresomes, implying an inability to fuse with lysosomes for destruction and recycling. Thus, salt 1-**3C** disrupts mitophagic flux. Utilization of glucose for production of energy and for synthesis of metabolic intermediates increased. When glucose (or pyruvate) are depleted to a level incompatible with maintenance of basal metabolism, salt 1-**3C** leads to overall metabolic collapse and cell death. mTOR = mammalian target of rapamycin.

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
