# Peer review of "A Cyclic Pentamethinium Salt Induces Cancer Cell Cytotoxicity through Mitochondrial Disintegration and Metabolic Collapse"

_ijms, 2019, doi:10.3390/ijms20174208_

Round 1

Reviewer 1 Report

  I was fascinated with their findings. The article, however, the authors need  to dig deeper and find the detail molecular mechanisms behind their findings.

  Previously, the same group had shown that quinoxaline 1-3C could arrest the growth of xenografted tumor in nude mice (shown in ref 14) with little adverse effect. In that article, the effect of 1-3C on tumor seems to be cytostatic, arresting tumor growth, but not cytotoxic, meaning the regression of the tumor. What happens to the tumor after the end of the treatment was not discussed; would they start to grow again, and if so, how soon? The distinction is important, especially since the autophagy could protect tumor cells from apoptosis (JCB 278 (36) 30664-30676),  making the tumor cells survive during the treatment, allowing tumor to regrow later. 

  In this article, Krejcir and colleagues tried to find the molecular mechanism of 1-3C induced growth arrest. Using cultured melanoma cell line A-375, they observed fragmentation of mitochondria within 15 minutes of 1-3C addition. Does mitochondrial transmembrane potential drops just as quickly? The possibility of using transmembrane indicator is discussed in the discussion, but they never show their findings. This is an easy assay. Why not do them? 

  Next, they tried to say that the autophagy is being induced, showing the loss of LC3 in 28 hours. However, the standard assays for autophagy examine LC3-I and LC3-II at much earlier time points, which is not done here. 

   Since they did not observe PARP cleavage in 24-48 hours, it is not apoptosis. i.e., caspase-dependent cell death. If they had added pan-caspase inhibitor z-VAD and prevented cell death, they can re-emphasize this point. The standard molecular model is that if mitochondria are engulfed in autophagosomes, they can not release cytochrome c, thus preventing apoptosis. 

   They also showed that oxygen consumption rate is reduced in 1-3C treated cells within 4 hours. It is important to  measure cellular ATP concentration, and see how quickly the ATP concentration drops. Is 1-3C enter mitochondria and directly interfering with electron transport chain, and if so, which chain in the complex is affected? All of them? Or 1-3C reduces cellular ATP concentration by some other ways, causing autophagy, thus interfering with mitochondrial function, including the release of cytochrome c.

   Lastly, the authors showed that viability of 1-3C treated cells were partially restored by addition of high glucose+pyruvate, but not in cells treated with low glucose+pyruvate.This all make sense, but the question is this: do 1-3C affects isolated mitochondrial? Because, then there is no autophagosome interfering with mitochondria in these experiments. 

   NIH3T3 cells grown to confluency  or inactivated HUMEC or HUVEC cells would be an excellent controls to show that 1-3C would not have any effect on normal quiescent cells. 

   Thus, do 1-3C arrest cell growth in all growing cells, but not necessary kill them? What is the threshold from growth arrest to cell death? Can we really maintain 1000 nM 1-3C in circulation for 72 hours, which is the condition used to kill cultured A375 melanoma cells (figure 7), in animals? These are important questions when we try to take study like this to the translational phase.

Author Response

I was fascinated with their findings. The article, however, the authors need  to dig deeper and find the detail molecular mechanisms behind their findings.

Previously, the same group had shown that quinoxaline 1-3C could arrest the growth of xenografted tumor in nude mice (shown in ref 14) with little adverse effect. In that article, the effect of 1-3C on tumor seems to be cytostatic, arresting tumor growth, but not cytotoxic, meaning the regression of the tumor. What happens to the tumor after the end of the treatment was not discussed; would they start to grow again, and if so, how soon? The distinction is important, especially since the autophagy could protect tumor cells from apoptosis (JCB 278 (36) 30664-30676), making the tumor cells survive during the treatment, allowing tumor to regrow later. 

In most cases we observed tumor growth arrest during treatment. In one experiment, we followed the tumors after the end of treatmen for another two weeks and observed slight tumor regrowth within 10 days. Such data might indicate that certain tumor cells survived, allowing tumor regrowth. Additional experiments would be needed to confirm this observation.

In this article, Krejcir and colleagues tried to find the molecular mechanism of 1-3C induced growth arrest. Using cultured melanoma cell line A-375, they observed fragmentation of mitochondria within 15 minutes of 1-3C addition. Does mitochondrial transmembrane potential drops just as quickly? The possibility of using transmembrane indicator is discussed in the discussion, but they never show their findings. This is an easy assay. Why not do them? 

Next, they tried to say that the autophagy is being induced, showing the loss of LC3 in 28 hours. However, the standard assays for autophagy examine LC3-I and LC3-II at much earlier time points, which is not done here. 

Since they did not observe PARP cleavage in 24-48 hours, it is not apoptosis. i.e., caspase-dependent cell death. If they had added pan-caspase inhibitor z-VAD and prevented cell death, they can re-emphasize this point. The standard molecular model is that if mitochondria are engulfed in autophagosomes, they can not release cytochrome c, thus preventing apoptosis. 

They also showed that oxygen consumption rate is reduced in 1-3C treated cells within 4 hours. It is important to  measure cellular ATP concentration, and see how quickly the ATP concentration drops. Is 1-3C enter mitochondria and directly interfering with electron transport chain, and if so, which chain in the complex is affected? All of them? Or 1-3C reduces cellular ATP concentration by some other ways, causing autophagy, thus interfering with mitochondrial function, including the release of cytochrome c.

Lastly, the authors showed that viability of 1-3C treated cells were partially restored by addition of high glucose+pyruvate, but not in cells treated with low glucose+pyruvate.This all make sense, but the question is this: do 1-3C affects isolated mitochondrial? Because, then there is no autophagosome interfering with mitochondria in these experiments. 

NIH3T3 cells grown to confluency  or inactivated HUMEC or HUVEC cells would be an excellent controls to show that 1-3C would not have any effect on normal quiescent cells. 

Thus, do 1-3C arrest cell growth in all growing cells, but not necessary kill them? What is the threshold from growth arrest to cell death? Can we really maintain 1000 nM 1-3C in circulation for 72 hours, which is the condition used to kill cultured A375 melanoma cells (figure 7), in animals? These are important questions when we try to take study like this to the translational phase.

We thank the reviewer for these insightful comments and thoughts. We agree that the suggested experiments will be valuable for the future, but are beyond the scope of our current manuscript and would take much longer than the 10 days allowed to revise the manuscript. In response to the comments, we have added these thoughts into the revised manuscript.

Reviewer 2 Report

The article "A cyclic pentamethinium salt induces cancer cell cytotoxicity through mitochondrial disintegration and metabolic collapse" shows interesting and solid results. The manuscript is easy to follow and the experiments are well-justified. 

I would only suggest the following minor changes before acceptation:

-Include the justification of the synthesis of the salt 1-8C. Why have the authors selected a chain of 8 carbons?

-The authors should justify the election of different concentrations of salt 1-8C compared to salt 1-3C. For example, 2000 nM of salt 1-3C for mitochondrial localization vs 5000 nM of salt 1-8C. 

-Concentrations indicated in the text for Figure 6 (lines 236 and 237) do not correspond to those in the graphs. In Figure 6, the authors show the effects of the salts from 5 to 0.15 uM and not only at 2000 nM for salt 1-3C or 5000 nM for salt 1-8C as indicated in lines 236 and 237. In addition, commas in the graph have to be substituted by dots, as dots are the decimal separators in English. 

-Please include OXPHOS in the abbreviation list. 

Author Response

I would only suggest the following minor changes before acceptation:

-Include the justification of the synthesis of the salt 1-8C. Why have the authors selected a chain of 8 carbons?

We chose salt 1-3C as a pilot model, which was developed and tested in our institute a few years ago. The compound showed selective mitochondrial localization with fast entry into cells. We found that the compound has interesting cytotoxic properties, so we selected it for this detailed study. We were looking for an appropriate model with salt 1-3C, and we decided for a compound with the same structural motif, but with increased lipophilicity, which can significantly affect its transport into the cell, localization and its toxicity. We chose pentamethinium salt with two octyl chains C8 on the side benzothiazole units as the model (salt 1-8C). The length of the C8 chain proved to be almost ideal for our studies, as longer chains caused complications in transferring to the aqueous phase and made the compound essentially unusable for our further cellular studies.

-The authors should justify the election of different concentrations of salt 1-8C compared to salt 1-3C. For example, 2000 nM of salt 1-3C for mitochondrial localization vs 5000 nM of salt 1-8C. 

The selection of different concentrations were due to different cytotoxicity, lipophilicity and optical properties of compounds (mentioned above). We have clarified the reasons for different doses in the revised manuscript.

-Concentrations indicated in the text for Figure 6 (lines 236 and 237) do not correspond to those in the graphs. In Figure 6, the authors show the effects of the salts from 5 to 0.15 uM and not only at 2000 nM for salt 1-3C or 5000 nM for salt 1-8C as indicated in lines 236 and 237. In addition, commas in the graph have to be substituted by dots, as dots are the decimal separators in English. 

-Please include OXPHOS in the abbreviation list. 

We apologies for these mistakes, which have been corrected in the revised manuscript

Reviewer 3 Report

The authors studied the cytotoxic mechanism of a previously described salt (1-3C) compared to a new analogue (1-8C), with an extended side carbon chain and demonstrated that salt 1- 3C, but not 1-8C, is rapidly incorporated into mitochondria, correlating with increased cytotoxicity of salt 1-3C. 

The manuscript is well written and the results interesting. However, since compound 1-8C was not reported before, the synthetic method should be reported together with spectroscopic data, at least 1H-NMR and 13C-NMR spectra. Moreover, the purity of the compound should be stated by performing HPLC or elemental analysis. Hence, major revisions are required.

Author Response

The authors studied the cytotoxic mechanism of a previously described salt (1-3C) compared to a new analogue (1-8C), with an extended side carbon chain and demonstrated that salt 1- 3C, but not 1-8C, is rapidly incorporated into mitochondria, correlating with increased cytotoxicity of salt 1-3C. 

The manuscript is well written and the results interesting. However, since compound 1-8C was not reported before, the synthetic method should be reported together with spectroscopic data, at least 1H-NMR and 13C-NMR spectra. Moreover, the purity of the compound should be stated by performing HPLC or elemental analysis. Hence, major revisions are required.

Thank you for the comments. We have included the requested information on synthesis of the compounds as supplementary information in the revised manuscript. Each substance was characterized by 1H, 13C and high resolution mass spectometry (HRMS), rather than HPLC. This information is also included. We hope these data are sufficient to the reviewer.

Round 2

Reviewer 1 Report

in the earler publication, ahe authors had already shown that 1-3C causes tumor growth arrest. The question now is if 1-3C is causing cell death or cell cycle arrest, and how.

In this article, they performed experiments that is clearly different from the standard autophagy assays. Since they already observed fragmentation of mitochondrial network in 15 minutes, they should be able to see LC-3 cleavage at about the same time. But they do not. If LC-3 is uploaded into mitochondria and reduces transmembrane potential, it is important to show this by measuring mitochondrial transmembrane potential- and this is relatively easy. It is also important to show which complex is interrupted by LC-3. And the easiest assay, measuring cellular ATP concentration, is not done. Why not?

Thus all together, they have very clear data suggesting apoptosis is not taking place (even they can be improved), and they have data suggesting autophagy may or may not be taking place.  And those are all they have. No molecular mechanism of how 1-3C is working. That is not very good. I have learned very little from this very interesting phenomenon.

Lastly, do just one experiment applying 1-3C on healthy cell!

Author Response

Response to Reviewer 1

in the earler publication, ahe authors had already shown that 1-3C causes tumor growth arrest. The question now is if 1-3C is causing cell death or cell cycle arrest, and how.

In this article, they performed experiments that is clearly different from the standard autophagy assays. Since they already observed fragmentation of mitochondrial network in 15 minutes, they should be able to see LC-3 cleavage at about the same time. But they do not. If LC-3 is uploaded into mitochondria and reduces transmembrane potential, it is important to show this by measuring mitochondrial transmembrane potential- and this is relatively easy. It is also important to show which complex is interrupted by LC-3. And the easiest assay, measuring cellular ATP concentration, is not done. Why not?

We agree with the reviewer that we have not “dotted every i and crossed every t”, but our data on the mechanism of action are clear and will not be altered by each of the experimental approaches suggested here and previously. The conclusions may be modified, depending on results, but the overall conclusions that are given in the title will remain broadly the same – we have shown that salt 1-3C causes mitochondrial disintegration and metabolic collapse, which was not known before. Nonetheless, we agree that measuring mitochondrial membrane potential (MMP) is an independent back-up, and we have now performed these analyses for both salts. The data show a rapid and profound drop in MMP after salt 1-3C with a much slower and weaker effect of salt 1-8C, helping to provide independent evidence. The method has been added to the revised manuscript and the data included in the revised manuscript (panels d and e of figure 2).

Regarding autophagy, we apologize for the wording of these data: We had written that “mitochondrial fragmentation is accompanied by autophagy” as the title for this Section, which is not an accurate statement and is not how we interpret the data – we apologize for the clumsy wording and thank the reviewer for drawing the issue to our attention. We had performed the earlier time points requested, which show no changes compared to control at these earlier times – we have added these findings to the revised manuscript (data not shown, since there is nothing to see) and we have altered the title of this section to ensure that our findings are accurately understood – we do not say that this is a direct effect, only that there is an effect. Thus, the data show that salt 1-3C causes alterations to the autophagic machinery but without evidence of induction of apoptosis, important findings for the cellular effects of this compound.

Thus all together, they have very clear data suggesting apoptosis is not taking place (even they can be improved), and they have data suggesting autophagy may or may not be taking place.  And those are all they have. No molecular mechanism of how 1-3C is working. That is not very good. I have learned very little from this very interesting phenomenon.

We do not agree – we have shown many aspects of the effects of Salt 1-3C on mitochondrial morphology (and now on MMP), on glucose/pyruvate dependency, alterations in mTOR/AMPK, alterations in autophagy-related proteins / lack of alterations in apoptotic pathway proteins, and effects on oxygen consumption rates. None of these data were previously known and have provided a valuable insight not only for Salt 1-3C mechanism of action but also potentially for the development of compounds that target mitochondrial vulnerability in cancer, including considerations of side chains and their optimization.

Lastly, do just one experiment applying 1-3C on healthy cell!

We apologize to the reviewer for not pointing this out more explicitly in our manuscript: Previous experiments had studied the effects of salt 1-3C on normal cells, along with a range of cancer cell lines. These data are already published and showed a massive difference in IC50 values between normal and tumor cells. Thus, it is already known that salt 1-3C is highly selective for cancer cells in culture and there is no reason to repeat and re-publish these data. To address the reviewer’s comment, we have added an explicit statement of the differences in sensitivity between transformed and non-transformed cells to the revised manuscript to clarify this issue. We have also added an explicit statement to support the relative lack of effect on normal cells in vivo in the previous xenograft experiments.

Reviewer 3 Report

Now, in my opinion, the manuscript deserves to be published in IJMS

Round 3

Reviewer 1 Report

Authors seem to be reluctant to do additional experiments to address the molecular mechanism of 1-3C induced loss of mitochndrial respiration (they have performed only one out of many suggested experiments).

Mitochondrial respiration inhibitor such as CCCP induces mitochondrial fragmentation, loss of transmembrane potential (TMP), and activation of autophage (cleavage oif LC-3 from I to II) quickly, usually within 15 minutes. But with the only additional experiment they performed, they only see mito fragmentation but not loss of TMP, nor LC-3 cleavage. Only after 24 hours, they observed loss of TMP.  Thus it seems unlikely that 1-3C is directly inhibiting mito respiration complexes I, II, or III, as they suggested in the discussion.

If this is indeed the case, 1-3C maybe directly affecting mitochondrial fussion/fission machinery without affecting mito respiration, which would be very interesting. Thus they need to examine components of fusion/fission machineries such as OPA1, MFN1/2, DRP1. Only then, we can say that they took a step towards deciphering how 1-3C works. NOTE: Opa1 has six isoforms and loss of longer isoforms can be observed easily within 15 min of cccp addition.Thus they can use cccp as a control, and test if their compound 1-3C could do this. Activations of PINK and Parkin may not take place because loss of TMP usually activate them. Thus if they see loss of long isoforms of OPA1 without PINK Parkin activation, then we are looking at genuinly new phenomenum; that forced fragmented mitochondria is causing loss of functional mitochondria in 24 hours, slowing cell growth. They can try using one of mitochondria fusion inhibitors for a control.

Of course, they should perfom at least one experiment applying 1-3C to non-transformed (non-cancerous) cells such as HUVEC, HUMEC, NIH3T3, etc.

Author Response

Many thanks for sending the third set of comments from Reviewer 1 of our manuscript, requesting yet another set of additional experiments that they would like us to perform. The reviewer is correct that those studies will be informative, but these experiments will take 6-9 months of work, at considerable expense. This is a separate study.

I would like to reiterate the main findings of our work: we set out to investigate the mechanism of action of a novel compound that has already been shown to be highly selective for cancer cell lines over non-transformed cells in vitro and, more importantly, does not show detrimental side-effects in vivo – therefore we do not need to use normal cells in vitro (the list of which keeps growing!) in our study – those experiments will not answer the question we asked – how does it work to disrupt cancer cell growth?

Our current data are comprehensive including dose and time response data for numerous end points, and we have also developed and investigated a “control” compound, providing additional chemical biology information useful for drug design. Our biological experiments show an early effect on mitochondrial morphology, with evidence of fragmentation. The reviewer had asked whether this is associated with changes in function, which we have now provided additional data into. We were able to provide these data very rapidly, because we had already started those experiments before receiving the second round of review comments. The reviewer incorrectly states that there is not an early drop in TMP – the data show an early fall in TMP, which progresses over time. Indeed, mitochondrial fragmentation shows an early but incomplete effect, which progresses over time. We have also studied a variety of aspects of mitochondrial function and used glucose/pyruvate manipulations and mTOR/AMPK measurements etc to demonstrate energetics as a major contributor to the effects of the drug. We also show that apoptosis in not involved, whereas autophagy is involved. We do not know exactly how autophagy is activated. Autophagy is obviously not a direct target, from the data we present, and we have never claimed that it was. In the response to the reviewer ‘s comment at round 2 of revision, we have included data (that we already had) to clarify their comment.

In themselves, these data are valid and important for understanding the mechanism of action of this compound, and therefore for this class of compounds, as well as for further our understanding of mitochondrial functions and vulnerabilities in cancer cells. As such, we feel that the data are highly publishable without yet more additional data.